# Study on Permeability Characteristics of Porous Transparent Gels Based on Synthetic Materials

**DOI:** 10.3390/polym13224009

**Published:** 2021-11-19

**Authors:** Gaoliang Tao, Qingshi Luo, Shaoping Huang, Yi Li, Zhe Huang, Zhijia Wu, Fan Zhang, Heming Dong

**Affiliations:** 1School of Civil, Architectural & Environmental Engineering, Hubei University of Technology, Wuhan 430068, China; tgl1979@126.com (G.T.); 15872575073@163.com (Q.L.); yi.li10566@outlook.com (Y.L.); hz102010822@163.com (Z.H.); wzj2301214319@163.com (Z.W.); 102010839@hbut.edu.cn (F.Z.); 2School of Intelligent Construction, Wuchang University of Technology, Wuhan 430223, China; 3Shiyan City Transportation Investment Co., Ltd., Shiyan 442000, China; dhm.wayne@hotmail.com

**Keywords:** transparent gel, porous polymer, permeability, seepage observation

## Abstract

Advanced knowledge of the permeability characteristics of transparent gels play a key role in providing a rational basis for the study of porous polymer permeability and the research on the migration behavior of superpolymer solutions. Thus, a new type of transparent gel was prepared to simulate porous media, with aim to observe and analyze the permeability characteristics of transparent gel under the conditions of our experimental design by combining a transparent soil test and simple particle image velocimetry. The experimental results showed that the permeability of the transparent gel was similar to that through actual soil. The permeability coefficients of the transparent gel under different pressure gradients varied greatly early in the experimental cycle, while changing only slightly afterward, showing an overall trend of decreasing first and then stabilizing. With the increase of the mass ratio, the permeability coefficient of the sample decreased, the distribution of the low-velocity zone of the intercepted section became wider and tended to move upward. Differences in spatial position also caused different patterns of velocity and direction. The findings presented in this paper contribute to providing a new direction for the study of porous polymer permeability and the porous migration of superpolymer solutions.

## 1. Introduction

Porous media have widely existed in life, and the percolation in porous media is one of the hot issues of current research, such as the flow law of an oil-repellent polymer solution in porous media in oil and gas extraction [1], the design of optimal polymer dosing when using polymer to cure calcareous sand in geotechnical engineering [2,3], and the transport of contaminants in polymeric porous microspheres. Different liquid phases manifest different patterns when flowing through the same porous media, but they also have certain similarities. Soils are naturally porous media, which have a wide variety of structures due being subject to natural conditions. It is important, then, to explore the principles of water’s migration in different soil structures in studying seepage through porous media, as current theories of soil seepage are more complete and theoretically reliable. Soils can be divided into saturated and unsaturated soils according to their water content. In most cases, natural soils will interchange between unsaturated and saturated states, in which the migration rates and flow paths of water are difficult to predict, and, thus, many scholars have explored and studied them with the help of experimental instruments and theoretical formulas. However, it is difficult to visualize the infiltration process inside the soil for real-time observation; therefore, such study is limited by the manufacture and selection of experimental equipment. 

In order to further explore the microscopic seepage of soil, some scholars have applied CT (Computed Tomography), MRI (Magnetic Resonance Imaging), and X-ray technologies in geotechnical fields [4,5]. However, due to the high cost and difficulty of experiments therewith, the application of these methods is unlikely to become popular. In addition, some scholars have predicted parameters, such as permeability coefficients and deformability of soils, by establishing mathematical models and theoretical derivations; but there are still unavoidable deviations between theory and practice [6,7,8]. Thus, the experimental technique of transparent soil was proposed, with the aim of improving the drawbacks of the above methods to some extent. Transparent soil is not a soil in the traditional sense, but is a synthetic substance—a transparent gel with similar properties to natural rock and soil. Its advantage over ordinary soils is that the seepage process can be observed and the seepage velocity at various locations within the soil can be monitored in real time.

To date, the research on transparent soil materials is mainly focused on the geotechnical field [9], but its advantages, of intuitiveness, non-perturbation, methodological convenience, and low cost could also play an irreplaceable role in the study of percolation in porous media. The current studies on the migration of the liquid phase of porous polymers and the percolation characteristics of polymer solutions mostly compare percolation volume and permeability coefficients from the macroscopic point of view, but there are fewer studies on the migration velocity and migration path of the liquid phase, while the use of transparent gel material (hereafter referred to as transparent soil), combined with particle image velocimetry (PIV), allows observation of the percolation process, which offers promising research prospects.

The study of transparent soil firstly originated in 1982, when Allersma [10] studied its stress–strain properties using crushed glass as the material. In 1990, Mannheimer [11] synthesized a transparent mud with amorphous silica powder and matching fluid, and observed the flow properties of the transparent mud. Since then, Mannheimer, Iskander, and Oswald investigated the consolidation and settlement of the synthesized transparent mud and found that its physical and mechanical properties were close to those of natural rocky soil, which also provided strong evidence supporting the experimental method of simulating natural rocky soil with transparent soil [12,13]. Conventional transparent soils, depending on their aggregates and pore fluids, can be broadly classified into the following types.

The first is a transparent soil synthesized with amorphous silica as aggregate and a calcium bromide solution or mineral oil mixture as pore fluid. It is often used to simulate clay because its physical properties (e.g., permeability, consolidation, and strength) are similar to clayey soils. Lehane and Gill [14] used the transparent soil technique to quantify the soil deformation during penetrometer installation. Hird et al. [15] studied pile permeability using transparent soil synthesized from silica and pore fluid. In addition, transparent soils synthesized with silica as aggregate were applied to sandy soil foundation improvement [16] and high-gravity centrifugal experiments [17].

The second type is a transparent soil synthesized by using silica gel as an aggregate and mineral oil or calcium bromide solution with a matching refractive index as a pore liquid. Sadek [18] has carried out a series of experiments showing that transparent clay, synthesized by silica gel, has similar physical properties to sandy clay, and thus it is usually used to simulate sandy clay. Iskander has reported that silica gel particles have the static and dynamic behavior of sandy clay [19,20].

The third type of transparent soil is a hydrogel named KI-GEL201 K-F2(Abuabeads). This hydrogel has the same refractive index as water, so the hydrogel can be studied experimentally with water as the pore fluid. Kazunori Tabe [21,22,23] made a transparent porous medium using the Aquabeads material and observed the seepage inside it. By visualizing the seepage inside the porous media, Kazunori Tabe concluded that this material is more suitable for the observation of 2D seepage processes and the study of contaminant migration processes inside natural soils.

The fourth type is a transparent soil made of fused silica sand as aggregate and filled with a certain proportion of mixed mineral oil. It has the advantages of low temperature sensitivity and low viscosity of pore liquid.

Laponite RD is a clay mineral containing magnesium, lithium and silicon, though its main component is lithium magnesium silicate, which is a white powder in its natural state, and, when added to water, it can form a gel containing a large amount of water-network structure, which has good thixotropy, dispersibility, suspension and thickening properties. The crystal structure unit of lithium magnesium silicate gel is a tiny sheet with a thickness on the nanometer scale. The surface of the flakes is covered with exchangeable cations, mainly Na^+^. When the gel particles are mixed with water, the water is adsorbed onto the surface of the flake in contact with Na^+^, which spreads the gel along with the flake, at which time the particles swell rapidly until the flake is separated. As the surface of the flake is negatively charged and the end surface is positively charged, the separated end surface of the flake is attracted to the surface of the other flake, thus rapidly forming a three-dimensional colloidal structure. This is more similar to the attraction pattern of natural soil to water molecules, so Laponite RD can be used to simulate natural soft clay. This material was first proposed by Alvarez and Mauricio [24] to simulate groundwater sediments, Gidley [25] formulated transparent clay using Laponite RD as the raw material to simulate marine soft clay, and observed the fluid movement paths and slip surfaces in the soil. Wallace and Rutherford [26] found that the properties of Laponite RD were similar to the Aquabeads, and Beemer and Aubeny [27] studied the trajectory of a towed anchor with this formulated transparent soil.

In this paper, Laponite RD was selected to make a transparent soft clay, and the infiltration test was conducted in combination with PIV, and the infiltration characteristics of this transparent clay were summarized.

## 2. Experimental Study on the Permeability Characteristics of Transparent Soft Clay

### 2.1. Preparation of Transparent Clay

Laponite RD powders (see in Appendix A) were synthesized by Guangdong Yuexin Chemical Co., and they were used for the preparation of transparent clay specimens in this study. These powder particles belong to the type of round flakes with a height of about 1 nm and a diameter of about 25 nm [25] and belong to a kind of flaky silicate with the chemical formula Li_2_Mg_2_O_9_Si_3_. Using a self-controlled distilled water machine to prepare distilled water.

Three groups (i.e., A, B, and C) of 500 g of transparent clay specimens, with different constituent ratios (Laponite RD powder/water = 15/485, 20/480, and 25/475) were prepared. In other words, the mass admixture ratio of Laponite RD powder in the three groups specimens was 3%, 4% and 5%, respectively.

Taking specimen A as an example, the transparent clay specimens were prepared by adding 485 g of distilled water in a beaker and pouring 15 g of Laponite RD powder into the beaker, slowly, 8–10 times while stirring the mixture with a magnetic mixer, and, after all the Laponite RD powder was added and there were no lumps in the suspension, it was placed in a vacuum pump for the evacuation test. After the specimens had completely formed a gel-like substance (see in Appendix Ab), the samples whose transparency met the experimental requirements were selected for ring-knife sampling. Two more groups of specimens, B and C, were prepared in a similar way.

### 2.2. GDS Flexible Wall-Permeation Test

(i)Introduction of the test instruments

Compared with the traditional test, the GDS flexible wall-permeation apparatus is a more advanced permeation apparatus that has good stability, high measurement accuracy, anti-corrosion and anti-pollution features, etc.

Figure 1 shows the permeameter with the flexible wall used in the test. A specimen with a diameter of 70 mm and a height of 20 mm was placed in the pressure chamber (specimen cross-sectional area 3846.5 mm^2^). A rubber membrane was used as a flexible wall in the experimental setup and was mounted on the side of the specimen. This effectively avoided sidewall flow during test loading and, thus, reduced the uncertainty of the experimental results. The test parameters were set through a computer system, and the test was controlled by three pressure/volume controllers. The top and the bottom of the chamber were connected to pressure sensors, which were used to measure confining pressure, pore water pressure, and back pressure (Figure 1).

(ii)Test procedure

Using a custom ring knife to take the sample, specimens were assembled according to Figure 2. The pressure chamber was fixed and filled with water. Then the air bubbles in the three controllers and the PVC pipe were removed, followed by setting the backpressure saturation for 48 h.

The pressure chamber was set to 35 kPa, and the difference between the top pressure and bottom pressure of the specimen was ΔP. The test ΔP increased from 1 kPa to 20 kPa step by step, and the test time is set to 12 h for each level of differential pressure. Three groups of specimens A, B, C were tested according to this method. The data on the computer were read and processed to obtain the permeability coefficients of the specimens under different pressure differences.

### 2.3. Test Results and Analysis

The data measured by the tests were compiled, and the line graphs of the permeability coefficients with time for the same mass ratio specimens at different osmotic pressure differences, and the line graphs of the permeability coefficients with time for different mass ratio specimens at the same osmotic pressure differences were plotted. The images of the three groups of specimens showed the same trend, Figure 3 and Figure 4 are used as examples for specific analysis in this paper.

From the above figures, it can be seen that all the specimens reflected a similar trend—that their permeability coefficients decreased over time, regardless of the permeability pressure difference, and the higher the Laponite RD content, the smaller the permeability coefficient of the specimen. There are three main reasons for this phenomenon: (i) the transparent clay specimens for the test were obtained by the ring-knife sampling method—compared with the traditional ring-knife clay specimens, no external loading compression process was carried out—such that some large pores had formed inside the transparent clay specimens. (ii) The cohesive force of the transparent clay material is small, and, moreover, slippage within the aggregate is easy to produce. Additionally, the porosity of the material decreases under the action of the infiltration pressure. (iii) The transparent clay specimen is not uniform at the beginning of the infiltration fashion; as a result, the larger pore channels within the specimen are gradually filled by the discrete colloids carried by the seepage flow with the increase of infiltration time, which results in a gradual decrease of the infiltration coefficient. After a period of time, the internal structure of the specimen tends to stabilize and the variation of the permeability coefficient decreases. Hence, when the seepage flow reaches stability, the permeability coefficients of transparent clay specimens under different permeability pressure differences are basically the same. It is noteworthy that the decay rate of the permeability coefficient of the specimens in the first 6 h is significantly higher than that at the other times. This indicates that this stabilization process of the internal structure is mainly concentrated in the first 6 h when infiltration occurs. In addition, the permeability coefficients of all three groups of specimens basically varied in the range of 10^−5^–10^−7^ cm/s, which is consistent with that of clay, as shown in Table 1, which verify the feasibility of the material as a clay simulant.

For further comparison and analysis, the line graphs of the variation of permeability coefficient with the different permeability pressures after stabilization of specimens in groups A, B and C were plotted and are shown in Figure 5. In addition, the permeability coefficients of different dry-density Hunan clays were obtained to compare with the permeability coefficients of specimens in groups A, B and C at the same level of permeability pressure, and the results were shows in Figure 5.

The above figure presents several patterns as follows:The permeability coefficient of the transparent clay specimens after stabilization without a large range of change, and the overall trend remained stable. Greater Laporite RD content corresponded to smaller fluctuations of the permeability coefficient, i.e., the more stable the pore structure of the transparent clay specimens. This is due to the aggregate support of transparent clay being weak, and, when the amount of Laporite RD is small, the spatial network structure formed is looser, leaving more large pores within the transparent clay, such that the aggregate support structure is not stable enough, leading to a fluctuating permeability coefficient. In addition, the greater the Laporite RD content, the smaller the transparent clay permeability coefficient. This is due to the fact that more Laporite RD can produce more colloidal material to fill the pores, so, the overall porosity of the material is lower and the permeability coefficient is smaller.The permeability coefficient of Hunan clay showed a trend of rising first and then becoming constant with the change of permeability pressure difference. This is because when the permeability pressure difference is small, the film water, in the fine internal clay pores, cannot flow until it has reached the critical hydraulic gradient, whereupon can seepage occur. The transparent clay material does not have this phenomenon, essentially because the gel formed by Laporite RD and water is electrically neutral and cannot attract water molecules to form a bound-water film, as occurs with clay particles. When the permeability pressure difference reaches the critical hydraulic gradient, the permeability coefficient of clay changes by the same law as that of transparent clay, that is, it remains constant.The permeability coefficient of Hunan clay decreases with the increase of the dry density after stabilization.

All of the findings mentioned above show that the transparent clay material made of Laporite RD is suitable for simulating the percolation of soft clay with smaller pores under a higher-than-critical hydraulic gradient. Also, an appropriate Laporite RD ratio should be selected, to suit the permeability characteristics of the simulated material.

## 3. Seepage Observation Test of Transparent Clay

### 3.1. Test Equipment and Basic Principle

In this paper, a Doc Martens bottle was applied as the seepage pressure loading device, as shown in Appendix A. The hydrostatic pressure of device permeation is determined by the height of the liquid level from the lower mouth of the glass tube to the outlet, so as long as the solution does not fall below the lower mouth of the glass tube, the increase or decrease of the solution on it will not affect the hydrostatic pressure inside the bottle, thus automatically maintaining a constant osmotic pressure and flow rate.

### 3.2. The Basic Principle of PIV Technology

The PIV technique allows the measurement of the velocity of a mass by measuring the displacement of the mass. Supposing there were a mass in motion in space, the equation of its motion is given by Equation (1):(1)$$\left\{\begin{array}{l} v_{x}\;=\;\frac{dx\left(t\right)}{dt}\;\approx \;\frac{x\left(t+\Delta t\right)-x\left(t\right)}{\Delta t}\\ v_{y}\;=\;\frac{dy\left(t\right)}{dt}\;\approx \;\frac{y\left(t+\Delta t\right)-y\left(t\right)}{\Delta t}\\ v_{z}\;=\;\frac{dz\left(t\right)}{dt}\;\approx \;\frac{z\left(t+\Delta t\right)-z\left(t\right)}{\Delta t} \end{array}\right.$$
where: *v_x_*, *v_y_*, *v_z_* are the velocities of the mass in the *x*, *y*, and *z* directions, respectively; *x*(*t*), *y*(*t*), and *z*(*t*) are the positions of the mass at time *t*; Δ*t* is the interval time; *x*(*t* + Δ*t*), *y*(*t* + Δ*t*), and *z*(*t* + Δ*t*) are the positions of the mass after an interval time.

A conventional digital image is composed of many pixels, composed in a rectangular arrangement. Each pixel corresponds to a specific value and code for grayscale, color, contrast, etc. Therefore, the pixel data of an image can be digitally described by arrays and matrices. Each image captured for this experiment was divided in a vertical and horizontal manner to form a grid, and each pixel in the grid was treated as a block, the brightness and darkness of which are reflected in a grayscale value for the pixel, as shown in Appendix A. The grayscale values range from 0 to 256, where a value of 0 indicates pure black and 256 indicates pure white, and tones therebetween are represented by other values within this range. Representing an image in terms of the grayscale values of its pixels, the grayscale values of points (*i*,*j*) can be represented by a two-dimensional discrete matrix *f* (*i*,*j*).
(2)$$f\left(i,j\right)\;=\;\left(\begin{array}{ccc} f\left(1,1\right) & K & f\left(1,M\right)\\ M & 0 & M\\ f\left(N,1\right) & L & f\left(1,1\right) \end{array}\right)$$
where *M* represents the number of pixels in the horizontal direction, *N* represents the number of pixels in the vertical direction, *f* (*i*,*j*) represents the grayscale value of the point (*i*,*j*).

Afterwards, the image correlation was processed and matching analysis Was performed, according to the grayscale values of the pixels in the image.
(3)$$C\left(\Delta x,\Delta y\right)\;=\;\iint_{A}I_{0}\left(x,y\right)I_{1}\left(x+\Delta x,y+\Delta y\right)dxdy$$
where *x* and *y* represent pixel coordinates, ∆*x* and ∆*y* represent displacement, *I*_0_ and *I*_1_ represent pixel values of images before and after displacement, and *C* is the correlation coefficient.

### 3.3. Test Procedure

Transparent clays with mass ratios of 3%, 4%, and 5% were prepared in custom-sized molds according to the method in Section 2.1. During the preparation of the specimens, a small amount of phosphor was added to be used as a percolation tracer.

After the test preparation was completed, the water outlet switch and laser light source were turned on in a dark environment to irradiate the cross-sections of the specimens, and the imaged cross-sections were shown in Appendix A. Two sections, namely K and M, were selected in the seepage area of each mass ratio specimen to compare the difference in flow rate between different sections. A high-definition camera was set up to take pictures at two-minute intervals during the test and grayed out. The PIVLab seepage analysis was performed using matlab software for two sections K, M of each mass ratio specimen, and the test procedure is shown in Figure 6.

### 3.4. Permeation Velocity Distribution Pattern of Specimens with Different Mass Ratios

The data analyzed by the software were organized, and the seepage velocity data points of the K and M sections were extracted. The magnitude of seepage velocity of the sections under each mass ratio is shown in Figure 7.

The results presented in Figure 7 clearly indicate that the distributions of seepage velocities in different sections was similar when the mass ratios were the same, while the dispersion of data points increased with increasing mass ratio. Comparing the seepage velocity plots of different mass ratio specimens at each point of the same section, the seepage velocity and infiltration velocity interval of 5% mass ratio specimens in the vertical direction was the smallest, and the seepage velocity in the horizontal direction, with the increase of mass ratio, shows the law of its increasing and decreasing.

### 3.5. Flow Velocity Clouds of Specimen Sections with Different Mass Ratios

In order to more intuitively reflect the distribution pattern of seepage flow velocity at different locations, the velocity clouds of each section of specimens with different mass ratios were drawn and are shown in Figure 8.

In the above figures, the flow velocity in the yellow area is faster, while the flow velocity in the blue area is slower, and the seepage direction is from right to left. It can be seen that the seepage in different sections of transparent clay of the same mass ratio is approximately the same, and the distribution of the low seepage-velocity area is also the same. However, there is a certain deviation of the flow velocities in the same area, which proves that the difference of spatial position in this test condition is responsible for the differences of flow-velocity magnitudes in different directions.

Comparing the seepage of transparent clay with various mass ratios, it can be seen that higher mass ratios corresponded to larger blue areas, more uniform distribution, and an overall low seepage velocity area is more widely distributed. Furthermore, the low seepage-velocity area tended to move upward.

Combined with the relevant conclusions in Section 2.3 and Section 3.4, the 4%-doping-ratio specimen had more points and larger percolation rates at each point, compared to the 3%-doping-ratio specimen; but the former specimen’s overall permeability coefficient was smaller because the overall permeability coefficient of porous media is more influenced by low flow-rate regions (smaller aperture). Therefore, the total permeability coefficient of the high-mass-ratio specimen was the smallest mainly due its having had the widest distribution of low flow-rate regions.

## 4. Conclusions

In this paper, the permeation properties of a synthetic porous material were investigated, and its transparent nature was leveraged to trace a water permeation process inside the material, providing new ideas for the study of permeation properties of porous media. The main conclusions of the article are as follows:

The permeability coefficients of specimens with different mass ratios were measured under different pressure differences, and the test results showed that the permeability coefficients of the specimens varied in the range of 10^−5^–10^−7^ cm/s, which was consistent with that of natural soft clay. During the test, the change of permeability coefficient was large in the first six hours. The permeability coefficient of the specimens in the stabilization stage decreased with the increase of the mass ratio.

Seepage observations were made on transparent clay specimens. The test results showed that with the increase of mass ratio, the low-velocity zone of seepage on the analyzed section of the specimen was more widely distributed and tended to move upward. The distribution of seepage velocities in different sections of the same mass-ratio specimens was similar, but the values were somewhat different.

In porous media with low permeability coefficients, the number of small-sized pores in the seepage path or the proportion of low-flow velocity regions have a significant influence on the permeability coefficient. The tests on the permeability characteristics of the examined transparent clay and the seepage observation tests thereof have verified the reliability of transparent clay material as an alternative material to soft clay, and provided a novel concept for the study of the permeability characteristics of porous media with similar soft clay structure.

## Figures and Tables

**Figure 1 polymers-13-04009-f001:**
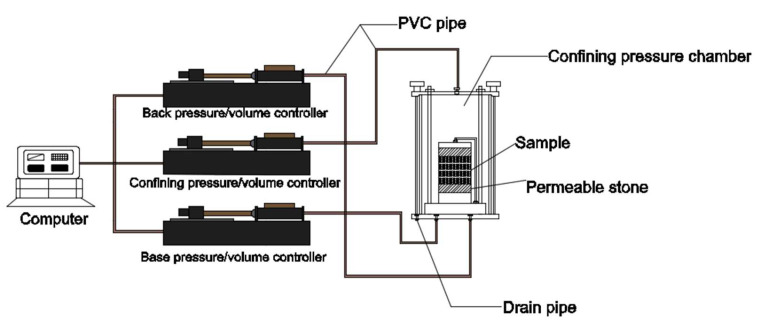
GDS flexible-wall penetrometer.

**Figure 2 polymers-13-04009-f002:**
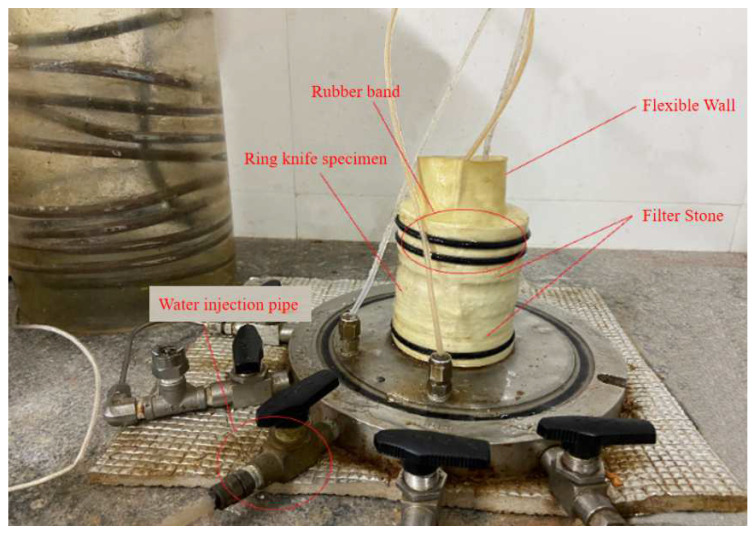
Physical picture of transparent clay permeability test sample assembly.

**Figure 3 polymers-13-04009-f003:**
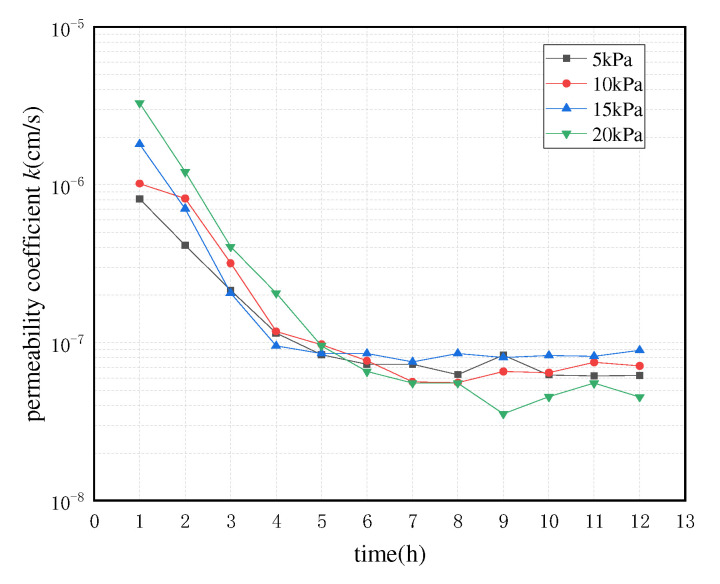
Folding line graph of the variation of permeability coefficient with time for Group C specimens under different permeability pressure differences.

**Figure 4 polymers-13-04009-f004:**
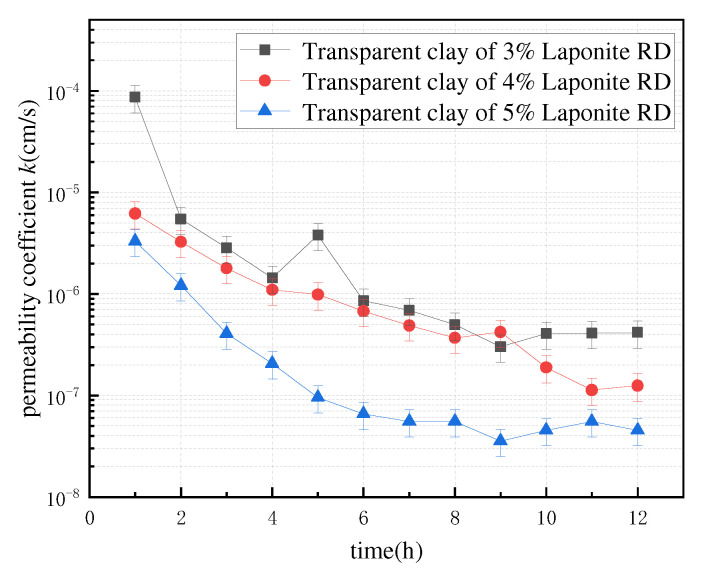
Folding line graph of osmotic coefficient with time for three groups of specimens at a 20-kPa osmotic pressure difference.

**Figure 5 polymers-13-04009-f005:**
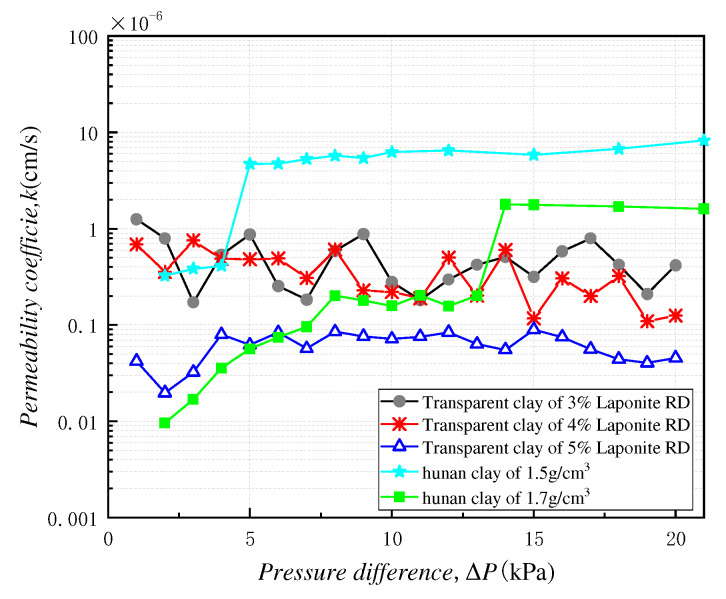
Line graph of the variation of permeability coefficient with the difference of permeability pressure for specimens with different mass mixing ratio.

**Figure 6 polymers-13-04009-f006:**
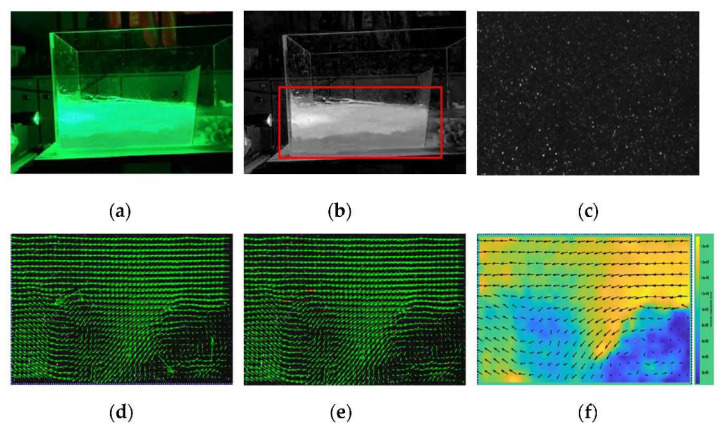
Test procedure: (**a**) physical picture; (**b**) grayscale processing; (**c**) processed images; (**d**) speed vector image; (**e**) vector calibration; (**f**) seepage velocity cloud.

**Figure 7 polymers-13-04009-f007:**
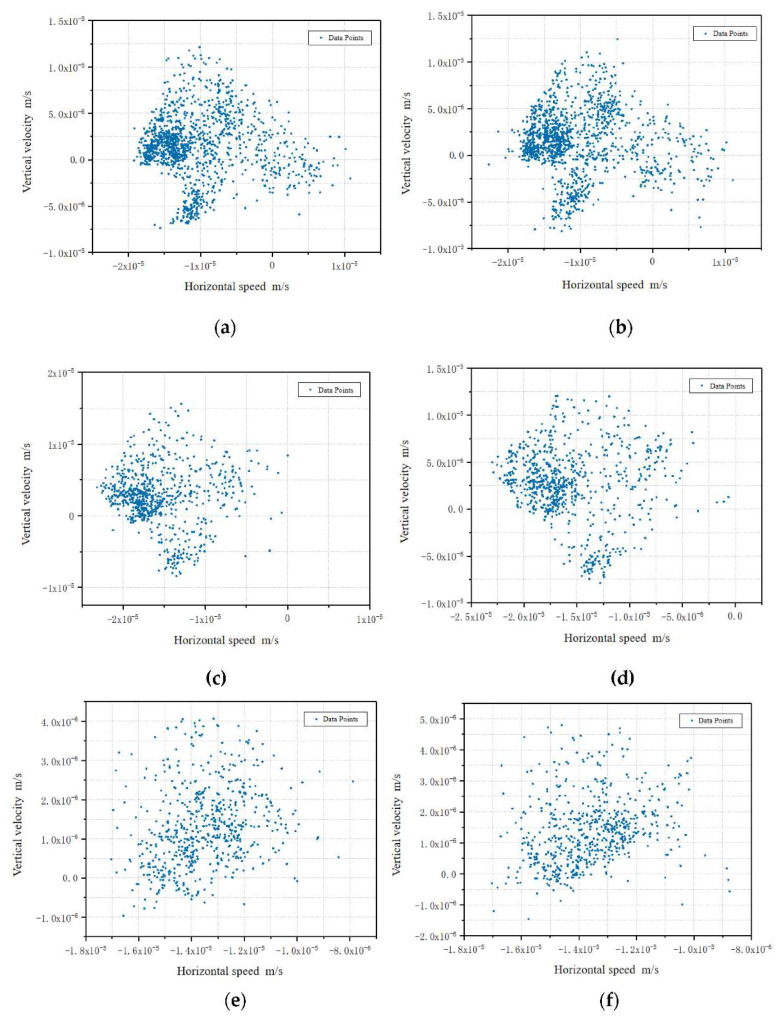
Comparison of seepage velocity of specimen sections with different mass ratios: (**a**) section K of a 3%-mass-ratio specimen; (**b**) section M of a 3%-mass ratio specimen; (**c**) section K of a 4%-mass-ratio specimen; (**d**) Section M of a 4%-mass-ratio specimen; (**e**) section K of a 5%-mass-ratio specimen; (**f**) section M of a 5%-mass-ratio specimen.

**Figure 8 polymers-13-04009-f008:**
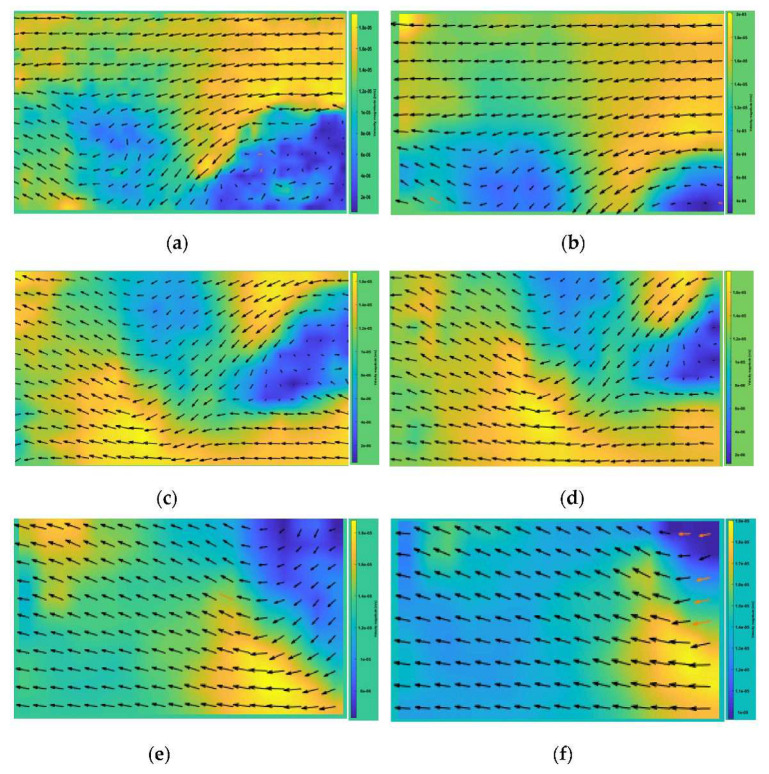
Flow velocity clouds of specimen sections with different mass ratios: (**a**) mass ratio of 3%—specimen K cross-sectional velocity cloud; (**b**) mass ratio of 3%—specimen M cross-sectional velocity cloud; (**c**) mass ratio of 4%—specimen K cross-sectional velocity cloud; (**d**) mass ratio 4%—specimen M cross-sectional velocity cloud; (**e**) mass ratio 5%—specimen K cross-sectional velocity cloud; (**f**) mass ratio 5%—specimen M cross-sectional velocity cloud.

**Table 1 polymers-13-04009-t001:** Comparison of saturated permeability coefficients of various types of clays.

Name of Soil Sample	Saturated Permeability Coefficient (×10^−7^ cm·s^−1^)	Data Source
Kga	1.421–5.000	reference [28]
Kga	0.750–3.100	reference [28]
Saz	0.140–0.990	reference [28]
Louiseville clay	0.169–1.460	reference [29]
Hunan clay	16.100–82.200	experimentally measured
Transparent clay A	1.720–12.500	experimentally measured
Transparent clay B	1.090–7.520	experimentally measured
Transparent clay C	0.197–0.837	experimentally measured

## Data Availability

Data is contained within the article or Appendix A. The data presented in this study are available in [insert article or Appendix A here].

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
