# Peer review of "Study on Permeability Characteristics of Porous Transparent Gels Based on Synthetic Materials"

_polymers, 2021, doi:10.3390/polym13224009_

Round 1

Reviewer 1 Report

The manuscript "Study on permeability characteristics of porous transparent gels based on synthetic materials" shows the use of Laponite RD a synthetic clay to analyze the permeability characteristics of transparent gel.  On one hand this might be interesting where Laponite RD is applied but as authors said to compare to natural clay there should be some comparison shown. It would be interesting so far if those permeability found as well in natural clay.

The authors said in abstract " The experimental results showed that the permeability of transparent gel are similar to that of actual soil". But there no data nor any references shown where such can be compared. Therefore if saying such better comparison either making them self or compare to other works has to be shown.

There two main issues in the script at first the vast amount of figures which don't help to make the manuscript better readable.

The 2nd part there no real scientific discussion provided no references to other work given in the result part. This has to be shown.

Some Figures are needed others can be placed in supplementary

Figure 1 just show image of Laponite RD powder that can be purchased so there no need show it.

In case for Figure 2 it just shows gel like material is formed if water added (can be shown in supplementary)

The real results are shown in Figure 5-Figure 7 where actually the discussion as well a proper description is missing. This is a vast amount of data as well there no standard deviation shown which means the reproducibility is not given. Please add standard deviations.

The actual results which are interesting shown in Figure 7 so as a suggestion for authors the Figure 5 and 6 can be moved to supplementary hence there no real explanation given what is the difference here.

In case of Figure 7 it would be actually more interesting adding natural clay to such to compare and see how much those synthetic clay match.

Figure 8 is actually experimental part and should be shown in such section same counts for Figure 9 (which doesn't give any information) can be moved to supplementary.

Figure 10 also belongs to experimental and the actual results are shown in Figure 11. So Figure 10 can be moved as well in supplementary.

Figure 12 it would be beneficial show in each Figure which test set used hence its quite tricky to separate the differences. Also in this case natural clay would match to find out if such perception valid saying that synthetic clay reflecting the natural case.

The studies have some novelty but need to be better presented more compact shown as well a proper discussion need to be added.

Reviewer 2 Report

The following comments should be considered while revising the manuscript before it can be recommended for the publication:

  • Highlight the potential applications of the transparent gel in the abstract.
  • In section 2.1, it is not clear “Laponite RD powder in the specimens were 3%, 4% and 5%, respectively” of what? Also, it is written “two groups of specimens B and C were prepared in a similar way as A”, does it mean samples A to C are the same? Then why samples are named them differently. Please clear it.
  • Fig 4 caption is incomplete. “Sample assembly physical picture” for what test? Please write it.
  • There are two figures with same numbering 5. Please fix it.
  • The procedure/formula for permeability coefficients values reported in Figures 5 to 7 are not discussed in methodology section properly.
  • It is not clear why the equations in section 3.2 numbered as 2.1? It is better to start from Eq. 1.
  • Is it possible to put the dimension of test chamber and distance between the laser device and chamber shown in Fig 10?
  • English should be improved as there are grammatical errors found in some sections. Please go through all the sections and correct them.

Round 2

Reviewer 1 Report

The authors answered all open questions and reorganized the manuscript. Therefore its now in publishable form.

Reviewer 2 Report

The authors have considered all the comments and revised the manuscript accordingly. I think now the manuscript is ready for publication.